# Entropy-Aided Meshing-Order Modulation Analysis for Wind Turbine Planetary Gear Weak Fault Detection under Variable Rotational Speed

**DOI:** 10.3390/e26050409

**Published:** 2024-05-08

**Authors:** Shaodan Zhi, Hengshan Wu, Haikuo Shen, Tianyang Wang, Hongfei Fu

**Affiliations:** 1Electronic and Control Engineering, School of Mechanical, Beijing Jiaotong University, Beijing 100091, China; sdzhi@bjtu.edu.cn (S.Z.); 21121251@bjtu.edu.cn (H.W.); shenhk@bjtu.edu.cn (H.S.);; 2Department of Mechanical Engineering, Tsinghua University, Beijing 100084, China

**Keywords:** fault diagnosis, meshing modulation, entropy, wind turbine planetary gear, entropy-aided meshing-order modulation

## Abstract

As one of the most vital energy conversation systems, the safe operation of wind turbines is very important; however, weak fault and time-varying speed may challenge the conventional monitoring strategies. Thus, an entropy-aided meshing-order modulation method is proposed for detecting the optimal frequency band, which contains the weak fault-related information. Specifically, the variable rotational frequency trend is first identified and extracted based on the time–frequency representation of the raw signal by constructing a novel scaling-basis local reassigning chirplet transform (SLRCT). A new entropy-aided meshing-order modulation (EMOM) indicator is then constructed to locate the most sensitive modulation frequency area according to the extracted fine speed trend with the help of order tracking technique. Finally, the raw vibration signal is bandpass filtered via the corresponding optimal frequency band with the highest EMOM indicator. The order components resulting from the weak fault can be highlighted to accomplish weak fault detection. The effectiveness of the proposed EMOM analysis-based method has been tested using the experimental data of three different gear fault types of different fault levels from a planetary test rig.

## 1. Introduction

Facing a severe energy crisis and environmental pollution, the energy conversation systems, such as the wind turbine, play a more and more important role in the modern industrial world. Hence, the stable and safe operation of the corresponding key subsystems, such as the planetary gearbox of the main drive train, is of great importance due to the harsh operational conditions, such as time-varying rotational speed and wind load. In fact, nonstationary operational conditions, especially the time-varying rotational speed, are common for the wind turbine, and will challenge the corresponding condition monitoring at the same time. As the key component in the wind turbine, the planetary gearbox is one of the fault sources which will affect the corresponding safety operation. Hence, it is quite vital to perform the corresponding condition monitoring and fault diagnosis under time-variant rotational speed according the wind stimulation. Therefore, accurate fault detection is necessary.

Several theoretical studies have been performed on this important topic, such as deriving the spectral complexity of measured vibration [1], exploring vibration transmission laws considering the transfer path effect [2], and revealing the spectral features resulting from a localized fault [3]. According to the conclusions derived from the aforementioned studies, two relatively difficult problems have been identified in this aspect; these problems include weak fault detection [4] and time-varying rotational speed [5].

Considering their complex structure and strong noise, it is difficult to identify prominent defect-related spectra from complex spectral clusters. The primary solution to the weak fault-related information enhancement problem can be summarized by exploiting the following three fault-related characteristics: periodicity [6], impulsiveness [7], and modulation [8]. Periodicity is common. Leveraging the periodicity and locating prominent defect-induced frequency components are the initial objectives. Considering the complex amplitude and frequency modulation phenomena, a conventional algorithm called time-synchronous averaging can be used to enhance the fault-related periodic information by eliminating unrelated components [6]. Several improved versions have recently been proposed to promote their extraction ability. In [9], the development of new window types and lengths was attempted. In [10], an expansion of time-synchronous averaging to multiple sensors was studied. In [11], the conventional version was improved to adapt to variable rotational speeds. Impulsiveness was the most straightforward feature. For this feature, several advanced impulsive component extraction methods, such as the spectral kurtosis [12], minimum entropy deconvolution [13], and stochastic resonance [14], have been used to highlight the fault features. However, early defects may not sufficiently stimulate strong impulsive components. Hence, it is difficult to identify the weak fault characteristics of the corresponding early stages. Amplitude and frequency modulation phenomena are the most fundamental characteristics of gear vibration signals. It is crucial to locate a modulated zone with as much fault information as possible. To address this issue, adaptive decomposition methods such as empirical mode decomposition [15], local mean decomposition [16], and variational mode decomposition [17] have been used to decompose raw signals, and the most relevant intrinsic mode function can be selected to highlight the fault feature. For weak fault detection, such as a defect on a planetary bearing, a meshing modulation indicator can be designed to identify the weak feature without using a healthy baseline [18]. Mesh modulation-induced methods have been proposed to detect ring gear fault features [19] from the demodulated spectra.

Time-varying rotational speed is a common operating mode for planetary gearboxes. Considering the complex spectral structure, capturing the time-varying fault-related time–frequency components is difficult, especially for weak faults with weak time–frequency features. To address this problem, two classical strategies exist: order tracking [20] and time–frequency analysis [21]. For the former, it is essential to acquire the accurate speed as prior information, and the fault features can be identified from the order domain. Another problem is that locating the modulated area under time-varying operating conditions challenges conventional methods. Regarding the latter method, the fault-related features can be located directly from the corresponding time–frequency representation (TFR) [22]; however, the weak fault features acting as multiple time-varying ridges with lower magnitude are difficult to identify using conventional time–frequency analysis methods because of the complex spectral distribution and weak faults.

Facing the difficulty mentioned above, the machine learning [23] or Bayesian approaches [24] may solve the problem to some extent. As for planetary gearbox fault detection, several deep learning methods have been employed, such as stacked autoencoder [25], deep brief network [26], convolutional neural network, transformer [27], generative adversarial network [28] and graph neural network [29]. These studies have enriched the planetary gearbox fault detection. However, the time-varying rotational speed also challenges the current deep learning-based method at some level. Hence, the study focuses on the conventional signal processing algorithm-based methods. 

In summary, although many studies have made significant progress in the detection of weak or time-varying fault features, the following difficulties have not been sufficiently addressed for cases where these two issues are combined. First, most modulation area detection methods are constructed to deal with the stationary operating mode. Although the demodulation analysis of particular modulation areas can highlight fault-related spectra, the corresponding performance is unsatisfactory under time-varying rotational speeds. Conventional strategies for the optimal modulation frequency band determination only consider the characteristic of impulsiveness, which is introduced by the localized fault. However, this kind of strategy is not suitable for detecting the weak fault, because it is not severe enough to simulate a high resonance response. Lastly, the target fault features under the time-varying rotational speed are generally one or two time-varying time–frequency ridges. First, these time-varying ridges may have low amplitude and the distance between different ridges is always low. Moreover, the slopes of all these components are proportional to each other. Even the most advanced time–frequency analysis algorithms, such as the scaling-basis chirplet transform (SBCT) [30] and velocity synchronous linear chirplet transform [31], have difficulty in capturing all of the time-varying components with relatively lower amplitudes that are close to those with higher magnitudes from the TFR of the raw signal. Therefore, the potential method should be able to detect the optimal weak fault-related frequency band under the time-varying rotational speed, and identify the target feature with less interruptions. 

To address these difficulties, an entropy-aided meshing-order modulation (EMOM) analysis-based method is proposed to realize the weak defect of a wind turbine planetary gear under time-varying operating conditions. For the proposed algorithm, an accurate time-varying rotational speed is first identified and extracted from a fine TFR with considerable resolution by employing a newly proposed scaling-basis local reassigning chirplet transform (SLRCT). In particular, a scaling basis is constructed to match the corresponding instantaneous frequency (IF) and obtain a precise TFR with high resolution. On this basis, a frequency reassignment operator is constructed using the principle of local maximum synchrosqueezing to extract time-varying speed information. Subsequently, an entropy-aided meshing modulation area detection method is constructed to detect the most sensitive area that modulates fault-related features. Specifically, the decomposition step of the conventional meshing modulation area identification method is first employed to separate the raw vibration, and then all the decomposed results are resampled in the order domain using the extracted rotational speed. Considering all the resampled separated results, the entropy-aided meshing-order indicator modulation (EMOM indicator) is designed to evaluate whether the meshing order is sufficiently prominent in the corresponding order spectrum. The area with the highest entropy-aided meshing-order indicator is selected as the most sensitive zone, and fault-related components are identified in the order spectrum of the filtered envelope of the raw signal with the meshing indicator lead frequency band. It is worth to mention that the final fault detection is realized based on the envelope spectrum in the order domain, rather than the TFR. It is much easier to locate the fault features from the spectrum than from the TFR, which makes the proposed algorithm suitable for the complex structure. To verify the effectiveness of this method, three different gear fault types are considered: “missing tooth”, representing the severe fault level, “tooth break”, representing the middle level, and “tooth root crack”, representing the weak level. It is worth mentioning that the fault types mentioned above cannot represent the severe, middle and weak levels in all the cases. However, compared with the missing tooth fault, the tooth break fault is weaker, and the tooth root crack fault is weaker than the tooth break fault. Hence, we set the missing tooth fault, the tooth break fault and the tooth root crack fault as the severe, middle and weak fault levels. Specifically, three experiments are conducted on a planetary gear test rig with a faulty sun gear. The main contributions of this study are as follows:(1)The strategy of the EMOM-based analysis is first expanded to the time-varying operational condition via the proposed novel EMOM analysis method.(2)The TFR, using the scaling basis, is refined using the local reassignment strategy, and a new SLRCT is constructed to realize accurate instantaneous rotational frequency extraction.(3)To identify the meshing modulation area under a time-varying rotational frequency, an EMOM indicator is designed, and an entropy-aided meshing-order gram (EMOMgram) is then constructed. Based on the EMOMgram, the frequency band with the highest indicator can be considered as the most sensitive, and the features of the weak gear fault can be located in the envelope order spectrum of the corresponding filtered result. With the help of information entropy, the selected optimal frequency band is relatively stable under different fault levels and operational modes.(4)We compare the proposed entropy-aided meshing modulation-based algorithm with the traditional Kurtogram-based algorithm for a planetary gearbox with three different fault levels on the sun gear. With the proposed method, fault-related features can be located not only in the lower orders, but also in the order representing the meshing frequency. This indicates that more features can be detected using the proposed algorithm. Furthermore, the proposed algorithm can locate the fault features of three different fault levels; however, the conventional method could only recognize severe gear faults of a missing tooth.

The remainder of this paper contains the following sections. Section 2 specifies the detailed steps of the proposed EMOM method. Section 3 demonstrates both the effectiveness and superiority of the proposed meshing-order modulation-based algorithm using vibration signals measured from a planetary gearbox test rig with three sun gear fault types at different fault levels. Finally, Section 4 concludes this study.

## 2. Entropy-Aided Meshing-Order Modulation Method

As described in Section 1, it is difficult to realize weak gear fault detection considering the background noise, time-variable rotational frequency, and complex structure. Hence, the new EMOM analysis method is proposed, in which the SLRCT-based IF extraction method is developed. Based on the extracted preview speed information, the conventional meshing modulation algorithm can be expanded to the condition of time-varying speed, in which a new EMOM indicator is designed to select the most sensitive frequency. Finally, the raw spectrum is filtered with the selected frequency band, and the corresponding weak gear fault-related features can be identified from the envelope spectrum in the order domain. The extracted fault feature appears not only in the lower order zone, but also around the peaks around the order representing the meshing frequency. The three parts of the proposed algorithm are described in the following subsections.

### 2.1. Instantaneous Rotational Frequency Extraction via Scaling-Basis Local Reassigning Chirplet Transform

For time-varying rotational speeds, the corresponding IF ridge is extracted from the TFR with a considerably high resolution. Specifically, the fine TFR is calculated using the SBCT. Subsequently, a frequency estimation operator is constructed according to the one-dimensional frequency reassignment principle. Finally, we combine the frequency estimation operator with the IF ridge detection tool to estimate the IF ridges of the rotational frequency completely and accurately.

First, for the vibration signal, *x*(*t*)∈*L*^2^(***R***), the corresponding TFR computed by the SBCT [30] can be expressed as follows:(1)$$SBCT_{x}\left(f_{c},t_{c}\right)=\int_{-\infty }^{+\infty }s\left(t\right)g\left(t-t_{c}\right)\exp\left(-j2\pi f_{c}\times \left(\begin{array}{l} t+\frac{\tan\left(\theta_{1}\right)}{m}{\left(t-t_{c}\right)}^{2}+\ldots \\ \frac{\tan\left(\theta_{1}\right)\cdot \tan\left(\theta_{2}\right)}{n}{\left(t-t_{c}\right)}^{3} \end{array}\right)\right)dt,$$
where *f_c_*∈***R*** represents the frequency center, *t_c_*∈***R*** represents the time center, *s*(*t*) represents the analytic signal, *θ*_1_, *θ*_2_∈(−*π*/2, *π*/2) represent the discrete rotation angles generated by orthogonal bases for the alternative TFR, *m* and *n* are constants, and *g*(*t*)∈*L*^2^(***R***) represents the normalized Gaussian window, which is expressed as follows:(2)$$g\left(t\right)=\frac{1}{\sqrt{2\pi }\sigma }\exp\left(-\frac{1}{2}{\left(\frac{t}{\sigma }\right)}^{2}\right),$$
where *σ* represents the standard deviation that determines the window width.

Second, according to the one-dimensional frequency reassignment principle in local maximum synchrosqueezing transform [32], the frequency reassignment operator for the TFR of the SBCT can be described as follows:(3)$$f_{r}\left(t_{c},f_{c}\right)=\left\{\begin{array}{ll} \underset{f_{c}}{\arg\max}\left|SBCT_{x}\left(f_{c},t_{c}\right)\right| & if\left|SBCT_{x}\left(f_{c},t_{c}\right)\right|\neq 0\\ 0, & if\left|SBCT_{x}\left(f_{c},t_{c}\right)\right|=0 \end{array}\right.,$$
where *f_c_*∈[*f_c_* − Δ, *f_c_* + Δ], and Δ represents the short frequency interval. The frequency reassignment operator is expressed as:(4)$$f_{r}\left(t_{c},f_{c}\right)=\left\{\begin{array}{ll} f_{i}\left(t_{c}\right), & if\;\;f_{c}\in \left[f_{i}\left(t_{c}\right)-\Delta ,f_{i}\left(t_{c}\right)+\Delta \right]\\ 0, & otherwise \end{array}\right.,$$

Therefore, the TFR calculated using the SBCT and, subsequently, the frequency estimation operator can be referred to as the local maximum scaling and reassignment operator (LMSRO), defined as:(5)$$LMSRO\left(f_{c},t_{c}\right)=\delta \left(f_{c}-f_{r}\left(t_{c},f_{c}\right)\right)=\left\{\begin{array}{cc} 1, & \left|f_{c}-f_{r}\left(t_{c},f_{c}\right)\right|<\varepsilon \\ 0, & otherwise \end{array}\right.,$$
where *δ*(•) denotes the Dirac delta function, and *ε* is a constant used to control the extraction of IF ridges.

Finally, to accurately extract the IF ridge and align it with the LMSRO, a tool for IF ridge [33] detection is introduced:(6)$$E\left(IF_{i}\left(t\right)\right)={\int_{-\infty }^{+\infty }\left|TFR\left(t,IF_{i}\left(t\right)\right)\right|}^{2}dt-\int_{-\infty }^{+\infty }\left(\lambda \cdot I{F^{\prime }}_{i}{\left(t\right)}^{2}+\beta \cdot I{F_{i}}^{\prime \prime }{\left(t\right)}^{2}\right)dt,$$
where the parameters *λ* and *β* adjust the regularization. The working process of this IF ridge detection tool can be understood as follows: (1) divide the TFR into *G* segments and determine the starting points for each time–frequency segment; (2) estimate the IF ridge considering the maximum amplitude in the corresponding TFR by employing the forward and backward methods based on time–frequency energy; (3) after obtaining the IF ridge of one modal component, set the corresponding amplitude to zero; and (4) substitute the remaining TFR into (7), and determine the next IF ridge trajectory through the above steps. Through the iterative process described above, all IF ridge trajectories can be detected.

To effectively determine the termination of the iteration process, the iteration criterion can be defined by substituting the LMSRO into (7) as follows: (7)$$\sum_{n=0}^{N_{f}-1}LMSRO\left(t_{n},IF\left(t_{n}\right)\right)=N_{f},$$
where *N_f_* represents the number of sampling points. However, in practice, errors inevitably occur in the estimation results. Therefore, it is possible to optimize the iteration criterion in (8), which is expressed as follows:(8)$$\sum_{n=0}^{N_{f}-1}SMRO\left(t_{n},IF\left(t_{n}\right)\right)<\rho N_{f},$$
where *ρ* is a constant [33], typically set to 0.8. By following the above procedure, we can accurately extract the IF ridges of the rotational frequency.

Considering that the issue of balancing the bias and variance trade-off for the time–frequency analysis [34] is vital, a discussion is given as follows: in time–frequency analysis (TFA), there are three sources of bias in the process. First, when using the SBCT to compute TFRs, it is important to accurately estimate the chirp rate. By considering the relationship between *a*(*t_c_*)·*f_c_* = tan(*θ*)∈(−∞, +∞) and choosing to calculate two rotation angles *θ*_1_, *θ*_2_(−*π*/2, *π*/2) using the tangent function with chirp rate, adjusting the number of rotation angles can improve the computational accuracy of the TFR as follows:(9)$$\left\{\begin{array}{c} \theta_{1}\left(i\right)=-\frac{\pi }{2}+\frac{\pi }{M+1}i\;\;\;\;\;\;\;\;i=1,2,3,\ldots ,M\\ \theta_{1}\left(j\right)=-\frac{\pi }{2}+\frac{\pi }{N+1}j\;\;\;\;\;\;\;\;j=1,2,3,\ldots ,N \end{array}\right.,$$
where *M* and *N* are the numbers of rotation angles.

Next, when using the local maximum scaling and reassignment operator (LMSRO) for fine-tuning the instantaneous frequency (IF) ridge, a ridge extraction threshold parameter ε is used to measure the deviation between the ideal IF and the computed IF, which can be described as follows:(10)$$LMSRO\left(f_{c},t_{c}\right)=\delta \left(f_{c}-f_{r}\left(t_{c},f_{c}\right)\right)=\left\{\begin{array}{cc} 1, & \left|f_{c}-f_{r}\left(t_{c},f_{c}\right)\right|<\varepsilon \\ 0, & otherwise \end{array}\right.,$$
Due to the calculation deviation between the frequency center and the frequency reassignment operator, the extraction of IF can be more accurate by setting a threshold.

Finally, for IF ridge detection, we utilize the principles of forward and backward estimation to establish a new iteration termination criterion using Equation (8), ensuring that the detected IF ridge is accurate enough within the allowable deviation range.

In conclusion, three biases occurred during the above TFA process, and they were all balanced using relevant thresholds and parameters.

### 2.2. Meshing Modulation Area Determination Using Entropy-Aided Meshing-Order Modulation Analysis

In this subsection, the most sensitive frequency band containing sufficient gear-fault information is determined using the proposed meshing-order modulation analysis method containing three steps, which are signal decomposition with filtering banks with a tree structure, EMOM indicator construction, and optimal frequency band determination according to a novel constructed EMOMgram. 

#### 2.2.1. Signal Decomposition

The raw signal is decomposed into several preset frequency bands to determine the most sensitive frequency band satisfying the meshing modulation area. In this study, the decomposition method used in [35] is employed, which includes the following two main steps:STEP 1: Filter banks in a binary tree structure containing multiple levels are first constructed by setting a low-pass prototype filter, and then designing quasi-analytic low-pass and high-pass filters using (11) based on the prototype in the following pyramidal manner:
(11)$$h_{0}(n)=h(n)e^{j\pi n/4},h_{1}(n)=h(n)e^{j3\pi n/4},$$
where *h*(*n*) is the prototype filter, and *h*_0_(*n*) and *h*_1_(*n*) are the corresponding low-pass and high-pass filters, respectively. Taking the *k*-level as an example, the number of sub-filtered bands is 2*^k^*.

STEP 2: The current filtered result is further expanded for higher resolution in a 1/3-binary manner.

#### 2.2.2. Entropy-Aided Meshing-Order Modulation Indicator

Taking advantage of the instantaneous rotational frequency extracted from Section 2.1, a novel indicator is constructed with the help of order tracking. First, all aforementioned filtered results are resampled into the order domain. Then, a novel indicator, namely the EMOM indicator, is constructed, as expressed by (12):(12)$$\begin{array}{ll} {\mathrm{Indicator}}_{EMOM} & ={\mathrm{Indicator}}_{MOM}/En(x_{o,en})\\ & =\sum_{i=1}^{3}\frac{1}{SeqPosition_{X_{o,en}}\underset{\left|o-io_{m}\right|\leq \varepsilon }{\max}X_{o,en}}/-\sum_{k=1}^{N}x_{o,en}(k)\times \log(x_{o,en}(k)) \end{array},$$
where EMOM indicates the entropy-aided meshing-order modulation, which can be classified into following two parts: Indicator*_MOM_* and *En*(*x_o_*_,*en*_). Indicator*_MOM_*_,_ which can be used to measure whether the orders representing the meshing frequency and its harmonics are prominent. In detail, *X_o_*_,*en*_ is the envelope spectrum of any filtered result in the order domain; *o_m_* is the order which represents the gear meshing frequency; *i* is the harmonic order, varying from 1 to 3; *ε* denotes the order tolerance, which is set as 0.5 in this study; $\underset{\left|o-io_{m}\right|\leq \varepsilon }{\max}X_{o,en}$ is the maxima value of *X_o_*_,*en*_ among the *i*th order range of |*o*-*io_m_*| ≤ ε; and $SeqPosition_{X_{o,en}}\underset{\left|o-io_{m}\right|\leq \varepsilon }{\max}X_{o,en}$ is the sequence position of $\underset{\left|o-io_{m}\right|\leq \varepsilon }{\max}X_{o,en}$ in *X_o_*_,*en*_. If prominent peaks exist in the order range of $\underset{\left|o-io_{m}\right|\leq \varepsilon }{\max}X_{o,en}$, then the corresponding sequence position is low, which leads to a higher value of Indicator*_MOM_*. Hence, this indicator aids in locating a sensitive meshing modulation range in the order domain. *En*(*x_o_*_,*en*_) is the information entropy of the envelope signal of the filtered result in the order domain. If a frequency band contains fault information, the entropy value will be relatively low. Hence, the Indicator*_MOM_*/*En*(*x_o_*_,*en*_) can not only reflect the modulation characteristic of the meshing order but also indicate the fault-induced impulsiveness.

#### 2.2.3. Entropy-Aided Meshing Modulation Area Determination via EMOMgram

By calculating the Indicator*_E__MOM_* of all the filtered results after performing the aforementioned signal decomposition step, an EMOMgram can be designed, and the one with the highest Indicator*_EMOM_* is considered to be the best candidate for the meshing modulation area.

### 2.3. Gear Weak Fault Detection Based on Filtered Order Spectrum

With the frequency band determined according to the EMOMgram, the raw signal is first bandpass filtered, and then resampled according to the instantaneous rotational frequency extracted from the TFR. Finally, the gear fault-related orders can be identified from the corresponding envelope spectrum.

## 3. Experimental Tests

### 3.1. Experimental Test Setup

The proposed EMOM method was verified using vibration signals with three different gear fault types—missing tooth, tooth break, and tooth crack—on the sun gear of the test rig in the Beijing Jiaotong University lab [23]. The experimental setup can be seen in Figure 1. In specific, two gearboxes, a planetary gearbox and a fixed shaft gearbox, are driven by the motor, which is controlled by an AC inventor. The corresponding parameters are listed in Table 1. In particular, the three types of gear-localized faults separately mimic severe, middle, and weak faults to verify the effectiveness of the proposed algorithm. After replacing the sun gear with faulty gears, as shown in Figure 2, an accelerometer was used to measure the corresponding vibration signals at different levels. In particular, the planetary meshing frequency *f_m_* was 21.875*f_s_^(r)^*, where *f_s_^(r)^* is the sun gear rotational frequency, and the corresponding fault characteristic frequency (FCF) of the sun gear (*fcf*_s_) was 3.125*f_s_^(r)^*. The sampling frequency was 48,000 Hz.

### 3.2. Fault Detection Results

#### 3.2.1. Missing Tooth Fault

In this subsection, the sun gear with missing tooth fault is mounted in the planetary gearbox to mimic the severe fault condition. In particular, a speed-up process was simulated by controlling the motor. The corresponding rotational frequency varied from 60 Hz to 25 Hz in a nonlinear way. The total duration of the measure vibration signal was 7 s. Figure 3 shows the corresponding vibration waveforms. It can be seen that the vibration severity becomes lower under the low rotational speed. Figure 4a shows the TFR result obtained using the proposed algorithm, and Figure 4b shows the extracted instantaneous rotational frequency. As shown Figure 4a, the time–frequency ridge, which represents the rotational, is prominent with amplitude superiority and less interruption. In addition, the resolution of the TFR was sufficiently high, such that the corresponding ridge representing the time-varying rotational speed could be correctly captured as shown in Figure 4b.

Figure 5a shows the conventional Kurtogram, and Figure 5b shows the EMOMgram obtained using the newly constructed algorithm. The most evident difference between these two figures is that the selected sensitive frequency bands differ. Considering that the sun gear has a missing tooth fault, the fault gear can stimulate the resonant frequency in a much higher frequency band at approximately 21,000 Hz with a much higher kurtosis value. According to the Indicator*_EMOM_* constructed in Section 2, the selected frequency center was 4000 Hz. According to these two different frequency bands, the raw signal was separately bandpass filtered. Figure 6 and Figure 7 show the envelope spectra of the filtered results in the order domain using the instantaneous rotational speed shown in Figure 4b.

For the analysis results using the conventional Kurtogram shown in Figure 6, there exist several significant fault spectral lines in the corresponding envelope spectrum in the order domain. In specific terms, the spectral peaks marked by *fcf_s_* and 2*fcf_s_* represent the FCF of the sun gear and its second harmonics. This can indicate the fault condition of the planetary sun gear. Besides these two peaks, the spectral lines representing the rotational frequency (*f_r_*) and the rotational frequency sidebands around the *fcf_s_* (marked by *fcf_s_* ± *f_r_*) can be detected easily at the same time. In summary, the Kurtogram can locate the missing tooth gear fault-related higher resonance frequency. As for the results of employing the proposed EMOM algorithm in Figure 7, the prominent peaks representing the FCF and the second harmonics of the sun gear fault can also be located in the low-order range. More importantly, the additional order peaks representing *f_m_* ± *fcf_s_*, and *f_m_* ± 2*fcf_s_*, which are also prominent, can be considered as evidence of the sun gear fault. By comparing these two envelope spectra in the order domain, the effectiveness of the proposed algorithm can be verified, and the corresponding superiority to the conventional Kurtogram is that more fault-related features exist in the EMOMgram-induced envelope spectrum.

#### 3.2.2. Tooth Break Fault

In this subsection, a weaker gear fault, “tooth break”, is considered, and the corresponding rotational frequency is varied from 37 to 56 Hz in 9.2 s. Figure 8 shows the raw signal waveform. Figure 9a,b show the TFR and extracted IF, respectively, which further indicate the effectiveness of the proposed speed information. Figure 10a,b show the Kurtogram and EMOMgram, respectively. For the Kurtogram, although the resonant frequency band with a center frequency of 21,750 Hz can be also located, the corresponding kurtosis value is much lower than that of the condition with a missing tooth fault. For the EMOMgram, a sensitive frequency band (center frequency: 3000 Hz, bandwidth: 6000 Hz) was located. 

Figure 11 shows the envelope spectrum of the filtered result of the measured signal with the sun gear tooth break fault and the frequency band determined by the Kurtogram. Different from the spectrum shown in Figure 6, only the prominent spectral peaks representing the rotational frequency and its second harmonic can be detected. No spectral lines representing the sun gear fault can be detected in the whole envelope spectrum in the order domain. The underlying reason is that the tooth break type of gear fault cannot lead to resonance in the higher frequency band as the missing tooth fault can. 

In comparison, Figure 12 displays the results obtained using the EMOMgram. At first, there exist prominent peaks at the spectral line representing the sun gear fault characteristic frequency (marked by *fcf_s_*). In addition, we can also locate the spectral peaks at the orders representing the rotational frequency sidebands around the *fcf_s_* (marked by *fcf_s_* ± *f_r_*). More importantly, the sideband beside the meshing frequency *f_m_* ± *fcf_s_* can also be identified as a localized peak. This result shows that the proposed EMOM algorithm is superior to conventional algorithms for identifying weak gear faults under the time-varying rotational frequency.

Comparing with the analyzed results of the gear missing tooth fault, although the frequency band (missing tooth fault: center frequency 21,000 Hz; gear tooth break fault: center frequency 21,750 Hz) selected by the Kurtogram are similar, there exists no evident fault-related feature for the tooth break fault. Similarly, the EMOMgram-based algorithm can also lead to similar optimal frequency bands under two different fault types (missing tooth fault: center frequency 4000 Hz; gear tooth break fault: center frequency 3000 Hz), however, evident fault-related features can be located from both two fault types as shown in Figure 7 and Figure 12. This shows the robustness of the proposed EMOM-based method. 

#### 3.2.3. Tooth Root Crack Fault

The weakest sun gear fault of the tooth root crack was used to verify the upper limit of the proposed algorithm. The corresponding rotational frequency was varied from 53 to 31 Hz in 4.1 s. Figure 13 shows the corresponding waveform of the raw vibration. Figure 14a shows the TFR using the SLRCT, and the corresponding IF is displayed in Figure 14b. It can be seen that the instantaneous rotational frequency can be easily extracted from the TFR with less interruption. Figure 15a,b show the Kurtogram and EMOMgram, respectively. As shown in Figure 15a, the crack fault cannot stimulate a higher resonance, as the obvious fault does, and the corresponding kurtosis value is lower. The center frequency and the bandwidth of the frequency band determined by the Kurtogram is 217,500 Hz and 1500 Hz. In addition, the center frequency and bandwidth of the target frequency band, determined according to the EMOMgram shown in Figure 15b, were 3000 and 6000 Hz. 

Figure 16 and Figure 17 show the corresponding envelopes of the filtered signals with the filter band determined using the Kurtogram and EMOMgram, respectively. In Figure 16, no evident spectral peak induced by the sun gear fault can be located. The only prominent order peak is the rotational frequency. For the envelope spectrum shown in Figure 17, the spectral peak at the order representing the sun gear FCF can be identified (*fcf_s_*), as well as the peaks beside the meshing frequency representing *f_m_* ± *fcf_s_*. This further demonstrates the effectiveness of the proposed algorithm in capturing subtle fault characteristics.

## 4. Conclusions

A new EMOM-based algorithm was constructed to realize weak gear fault detection under the operational condition of time-varying rotational speed with the following three main contributions: (1) extracting an accurate instantaneous rotational frequency with the newly proposed SLRCT; (2) identifying the meshing modulation area according to the newly constructed EMOM indicator under the time-varying rotational speed with the help of information entropy; and (3) highlighting the gear fault-related weak signature from the envelope spectrum in the order domain of the filtered result determined by the newly constructed EMOMgram. Three different gear fault types—missing tooth (severe fault), tooth break (middle fault), and tooth root crack (weak fault)—were added to the sun gear of a planetary gearbox test box to verify the effectiveness of the EMOM-based algorithm. The diagnostic performance of the proposed method indicates that the EMOM analysis can not only identify the fault-related features of severe gear faults, such as the missing tooth and the tooth break fault, but also detect the weak tooth root crack fault feature. In contrast, the conventional Kurtogram-based method can only detect a strong fault of a missing tooth, which clearly demonstrates the effectiveness of the new method in detecting weak gear faults under time-varying rotational speeds. In addition, the other superiority of the proposed algorithm is that the corresponding fault features can be detected not only in the lower frequency area but also beside the order representing the meshing frequency. We believe that this advantage will make the proposed approach more robust. 

## Figures and Tables

**Figure 1 entropy-26-00409-f001:**
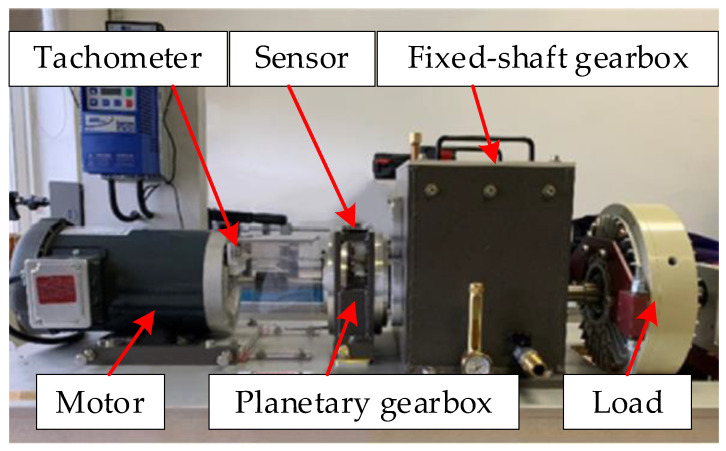
Lenze ESV222N02YXB Fault diagnosis test rig.

**Figure 2 entropy-26-00409-f002:**
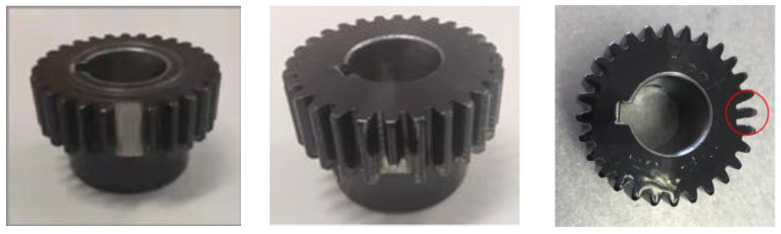
Sun gear failure modes (**left**: missing tooth; **middle**: tooth break; **right**: tooth crack).

**Figure 3 entropy-26-00409-f003:**
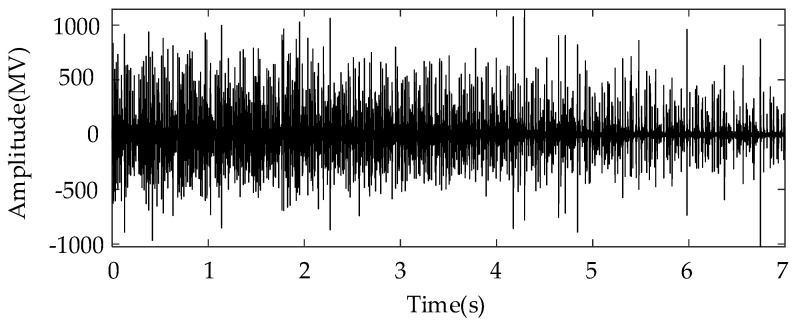
Raw signal waveform with gear missing tooth fault.

**Figure 4 entropy-26-00409-f004:**
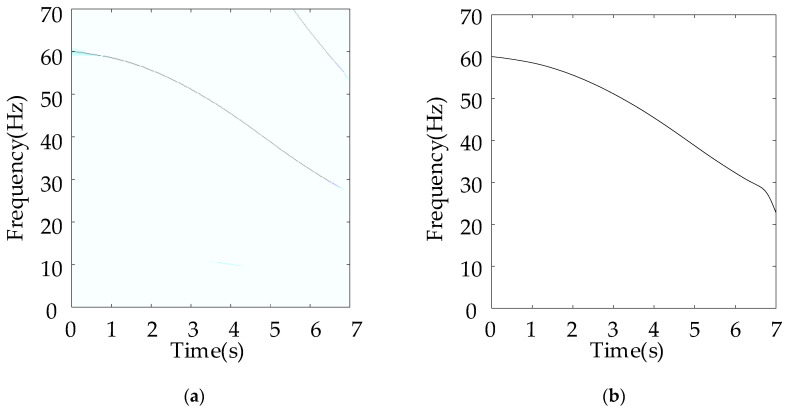
(**a**) TFR, and (**b**) instantaneous rotational frequency with sun gear missing tooth fault.

**Figure 5 entropy-26-00409-f005:**
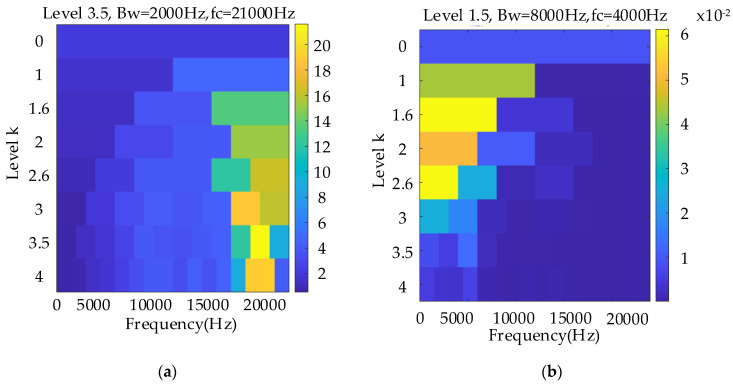
Kurtogram (**a**), and EMOMgram (**b**) of the raw signal with sun gear missing tooth fault.

**Figure 6 entropy-26-00409-f006:**
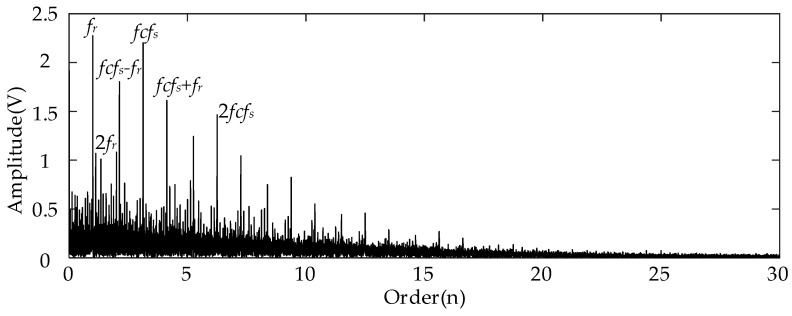
Envelope spectrum of filtered result with frequency band determined by Kurtogram with sun gear missing tooth fault.

**Figure 7 entropy-26-00409-f007:**
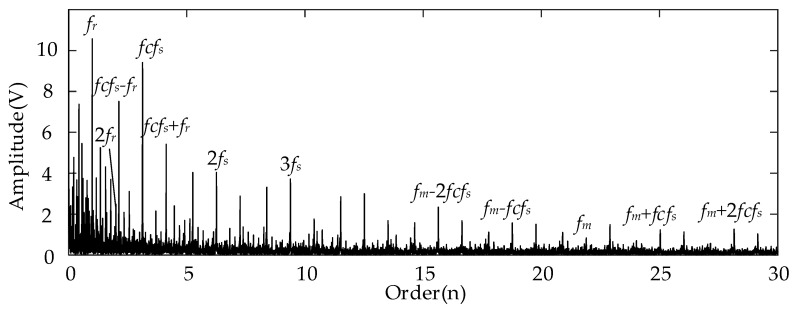
Envelope spectrum of filtered result with frequency band determined by EMOMgram with sun gear missing tooth fault.

**Figure 8 entropy-26-00409-f008:**
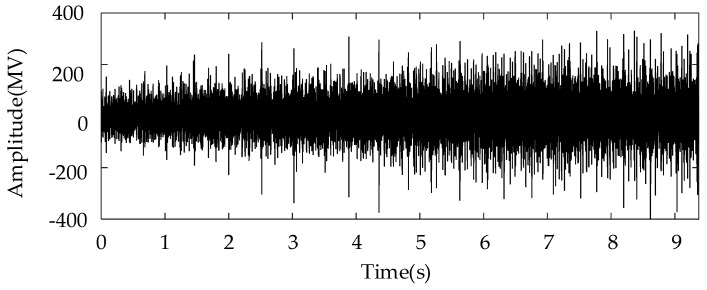
Raw signal waveform with gear break tooth fault.

**Figure 9 entropy-26-00409-f009:**
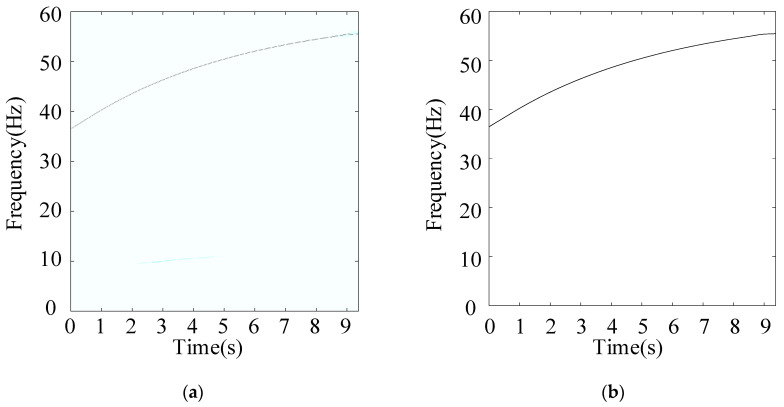
(**a**) TFR, and (**b**) instantaneous rotational frequency with sun gear tooth break fault.

**Figure 10 entropy-26-00409-f010:**
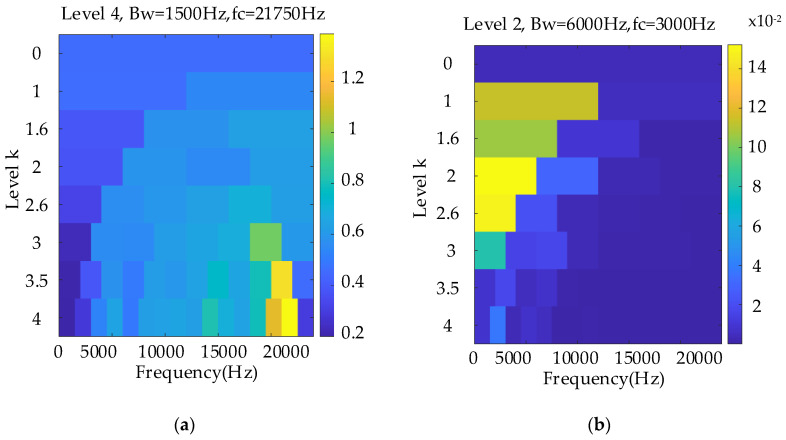
Kurtogram (**a**), and EMOMgram (**b**) of the raw signal with sun gear tooth break fault.

**Figure 11 entropy-26-00409-f011:**
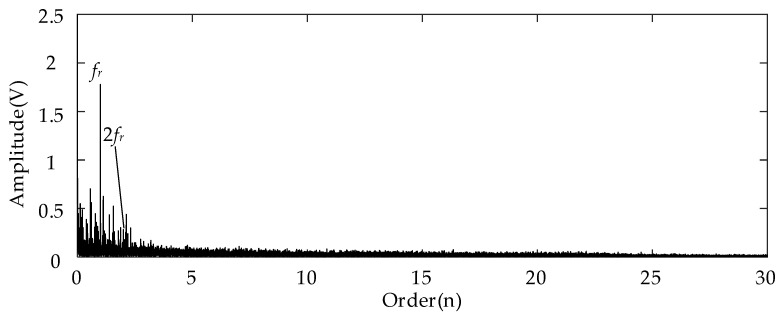
Envelope spectrum of filtered result with frequency band determined by Kurtogram with sun gear tooth break fault.

**Figure 12 entropy-26-00409-f012:**
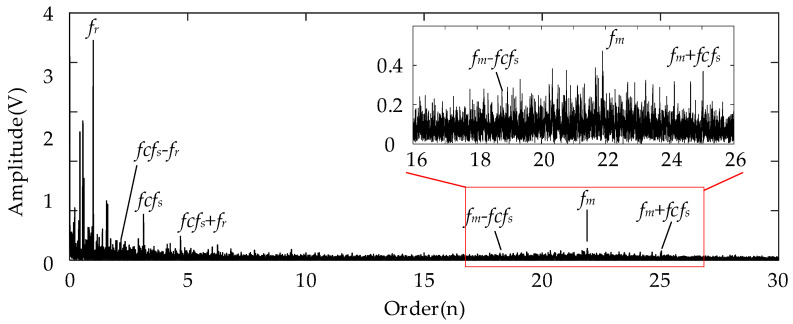
Envelope spectrum of filtered result with frequency band determined by EMOMgram with sun gear tooth break fault.

**Figure 13 entropy-26-00409-f013:**
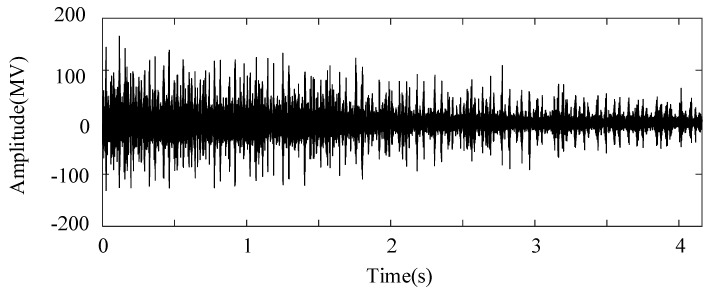
Raw signal waveform with gear tooth crack fault.

**Figure 14 entropy-26-00409-f014:**
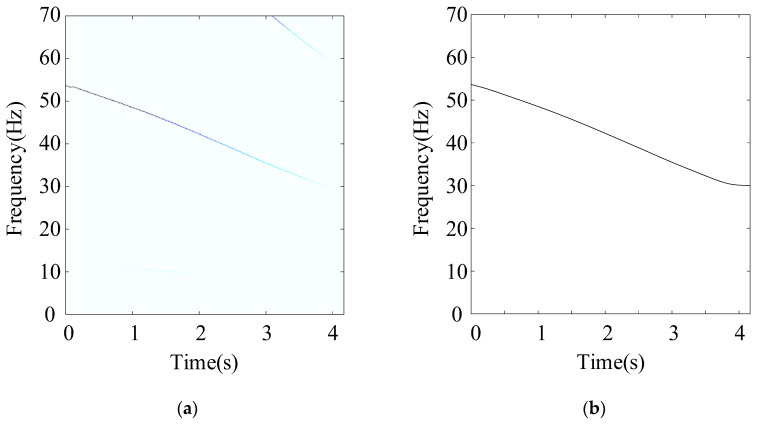
(**a**) TFR, and (**b**) instantaneous rotational frequency with sun gear tooth root crack fault.

**Figure 15 entropy-26-00409-f015:**
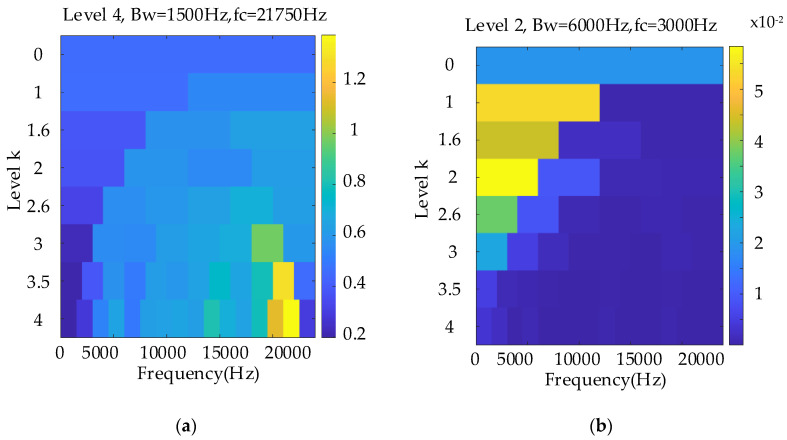
Kurtogram (**a**), and EMOMgram (**b**) of the raw signal with gear tooth crack fault.

**Figure 16 entropy-26-00409-f016:**
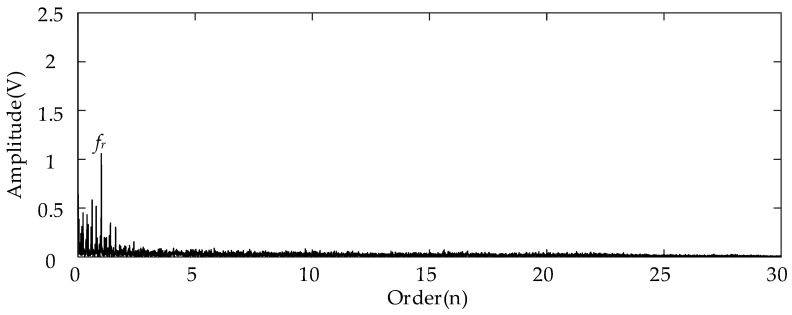
Envelope spectrum of filtered result with frequency band determined by Kurtogram with gear tooth crack fault.

**Figure 17 entropy-26-00409-f017:**
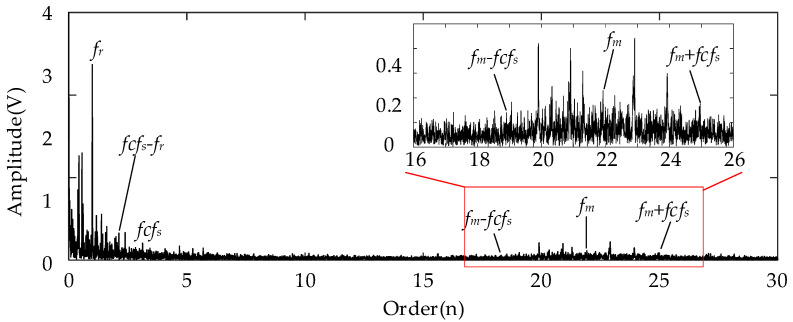
Envelope spectrum of filtered result with frequency band determined by EMOMgram with gear tooth crack fault.

**Table 1 entropy-26-00409-t001:** Parameters of planetary gearbox of Lenze ESV222N02YXB Fault diagnosis test rig.

Title 1	First Level			Second Level		Third Level	
Gear	Sun gear *Zs*	Planetary gear *Z_p_*	Ring gear *Z_r_*	Low speed *Z_2l_*	High speed *Z_2h_*	Low speed *Z_3l_*	High speed *Z_3h_*
Tooth number	28	36 (4)	100	100	29	90	36
Ratio	5.647			3.44		2.5	

## Data Availability

The data presented in this study are available on request from the corresponding author.

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
