# Peer review of "Entropy-Aided Meshing-Order Modulation Analysis for Wind Turbine Planetary Gear Weak Fault Detection under Variable Rotational Speed"

_entropy, 2024, doi:10.3390/e26050409_

Round 1

Reviewer 1 Report

Comments and Suggestions for Authors

This paper presents an entropy-aided meshing-order modulation method for weak gear fault detection of wind turbines. This method is illustrated in detail, and the paper is well-written. It is suggested to consider the following comments before it is published. 

  1. The Introduction should be improved. In practice, machine learning or Bayesian approaches are also popular in fault detection of structures, such as Switching Bayesian dynamic linear model for condition assessment of bridge expansion joints using structural health monitoring data.
  2. For the 2.2.1 section, ‘In this study, the decomposition method used in [33] is employed, which includes the following three main steps.’ However, only two steps were listed.
  3. It is not easy to detect faults based on the time-frequency analysis of experimental results. More detailed discussions should be added.
  4. How did you balance the bias and variance trade-off for the time-frequency analysis? It is suggested to refer to An improved multi-taper S-transform method to estimate evolutionary spectrum and time-varying coherence of nonstationary processes.
Comments on the Quality of English Language

 Minor editing of English language required.

Reviewer 2 Report

Comments and Suggestions for Authors

General comments:

This paper proposes a novel EMOM-based algorithm to detect weak gear faults under the operational condition of time-varying rotational speed. In the study, the authors developed a novel scaling-basis local reassigning chirplet transform (SLRCT) method to extract rotational frequency. In addition, they constructed a new entropy-aided meshing-order modulation (EMOM) indicator to identify the most sensitive modulation frequency area based on the extracted fine speed trend. Lastly, they filtered the raw vibration signal through the optimal frequency bank indicated by the EMOM. Three gear fault types under different fault severity levels were analyzed to demonstrate the effectiveness of the proposed method.     

The paper presents a novel method with a solid evaluation process. However, some illustrations are not very clear, making it hard to follow. I invite the authors to address a few comments below before accepting the manuscript. 

Technical comments:

1. How are fault severity levels defined?

2. The full name of the abbreviation of EMOMgram should be given.

3. Equation 7 should be explained. Specifically, Nf should be explained. 

4. Are there any references that can be cited for the setting of ρ in Equation 8? 

5. The illustration of the method in Section 2.2 should be better structured if the method contains several methods. The context (from Lines 210 to 225) does not provide a very overall picture of the method. It makes it confusing to follow the method.  

6. In Section 2.2.1, the authors stated there are three main steps (Lines 229 to 230). However, only two steps are presented. 

7. The Section numbers are wrong. There are two Section 3, and three Section 3.1.

8. I would suggest dividing the current Section 3 into two Sections as test setup and fault detection result to make it easier to follow. The illustration of the fault detection results should be improved. 

9. The description of the experiment is not sufficient. 

Editorial comments:

1. How are fault severity levels defined?

2. In line 242 in Section 2.2.2, the “where” in the explanation of Equation 10 should not be capitalized. 

3. The resolution of Figures 4(a), 9(a), 14(a) should be increased. It is suggested to deepen the line color. 

4. In line 306, line 313, and line 399, K in “kurtogram” should be capitalized.

5. The sentence from lines 95 to 97 should be re-written to make it clear. 

6. It is suggested to provide a nomenclature list as there are many abbreviations in the context.

Comments on the Quality of English Language

Overall, the English writing is good. But some sentences should  be improved to make it easier to follow.

Round 2

Reviewer 1 Report

Comments and Suggestions for Authors

The authors have addressed my concerns.

Comments on the Quality of English Language

Minor editing of English language required.

Reviewer 2 Report

Comments and Suggestions for Authors

The authors addressed my comments. Thanks!